# Epidemiological clustered characteristics of coronavirus disease 2019 (COVID-19) in three phases of transmission in Jilin Province, China

Qinglong Zhao[1,2◉], Yang Zhang[2◉], Meina Li[3◉], Rui Tian[2], Yifei Zhao[2], Bonan Cao[2], Laishun Yao[2], Xi Sheng[2], Yan Yu[1]*

**1** School of Public Health, Xi'an Jiaotong University Health Science Center, Xi'an Shaanxi, 710061, China, **2** Jilin Provincial Center for Disease Control and Prevention, Changchun, 130062, Jilin Province, People's Republic of China, **3** The First Hospital of Jilin University, Changchun, 130021, Jilin Province, People's Republic of China

◉ These authors contributed equally to this work.
* yuyan@mail.xjtu.edu.cn

**Data Availability Statement:** All relevant data are within the manuscript and its Supporting Information files.

## Abstract

The current epidemiological status of the new coronary pneumonia epidemic in China is being explored to prevent and control the localized dissemination of aggregated outbreaks. This study analyzed the characteristics of new outbreaks of coronavirus disease 2019 (COVID-19) at three stages of aggregated outbreaks in Jilin Province, China, to provide a reference for the prevention and control of aggregated outbreaks. Case information were collected from all patients in Jilin Province from January 12, 2020 to the present. The epidemic was divided into three stages according to the time of onset. The first stage comprised 97 cases reported from January 12, 2020 to February 19, 2020, during which 17 aggregated outbreaks occurred. The second comprised 43 cases reported from April 25, 2020 and May 23, 2020, involving one aggregated outbreak. The third comprised 435 cases reported on January 10, 2021 and February 9, 2021, involving one aggregated outbreak. The relationship between aggregated and non-aggregated cases in the first phase of the outbreak and the difference between imported and local cases during the aggregated outbreak were assess using statistical analysis, and the differences in the baseline information between the three phases were analyzed. The incubation periods of the three phases were 10 days, 8 days, and 5 days. The number of aggregated epidemic events in Jilin Province tended to increase and then decrease over time. The clustered events in Jilin Province were divided into four categories: household contact (14 times, 51 cases); household contact and public places (one time, three cases); household contact, public places, and gatherings (one time, six cases); and household contact, public places, gatherings, and work (three times, 495 cases). Clustered events occurred mainly between January 22, 2020, and February 4, 2020. Among all cases in the first phase of the outbreak, the method of detection and the time from diagnosis to discharge were longer in aggregated cases than in non-aggregated cases, and that the source of infection and renewal cases were more frequent and more likely to be detected in the outpatient clinics during aggregated outbreaks than the imported cases. The second phase of the epidemic showed significant spatial variability

**Funding:** The author(s) received no specific funding for this work.

**Competing interests:** The authors have declared that no competing interests exist.

(Moran's $I<0$, $P<0.05$). The third stage of the epidemic occurred in a higher proportion of individuals aged 50–90 years and within a shorter incubation period compared with the first two stages. The current focus of prevention and control of the COVID-19 epidemic in Jilin Province is to strictly implement the restrictions on gatherings and to perform timely screening and isolation of close contacts of infectious sources while strengthening the supervision of the inflow of people from outside the region. Simultaneously, more targeted prevention and control measures can be implemented for different age groups and occupations.

## Introduction

The coronavirus disease 2019 (COVID-19) has become a global pandemic [1]. On January 30, 2020, the World Health Organization (WHO) officially declared COVID-19 as a public health emergency of international concern. According to the WHO, the final number of patients with severe acute respiratory syndrome (SARS) infections in 2003 was 8,098, resulting in 774 deaths [2]. The final number of people infected with Middle East respiratory syndrome (MERS) was 2,220, resulting in 790 deaths [3]. Compared with SARS and MERS, COVID-19 is more widespread and is transmitted faster, with a more severe socioeconomic and health burden, resulting in 548,990,094 confirmed cases and 6,341,637 deaths worldwide [4,5]. No effective antiviral drugs have been developed for COVID-19, with isolation and symptomatic treatment being the mainstay of treatment. The COVID-19 vaccine has been widely administered worldwide, and some high-income countries have achieved a vaccination coverage of over 70% [6,7].

According to existing studies [8–11], the main modes of transmission of COVID-19 are respiratory droplets and close contact with an infected individual. The average incubation period of confirmed patients is 5 (2–14) days, and the most common clinical signs include fever, cough, and sputum production. COVID-19 is divided into four categories according to the severity of illness: mild, normal, severe, and critical pneumonia. In addition, asymptomatic infections are still considered infectious although they do not show any impact on the health of the organism [5,12,13]. This patient group was mainly identified through screening for nucleic acids in close contacts and secondary contacts of confirmed cases during a cluster outbreak. Asymptomatic infected individuals are less easily detected and identified, are more difficult to locate, and have more insidious course of disease transmission; therefore, they play a significant role in the spread of COVID-19 in the population.

The current local outbreaks of COVID-19 in China are mainly due to exposure to infected individuals while attending large gatherings, staying in public places, and under inadequate personal protection conditions. Due to the large number of people involved, susceptible populations may come from different regions and cause outbreaks in their own communities when they are infected. If infected individuals are not detected and effectively controlled in a timely manner, multiple outbreaks may occur, thus posing a significant public health challenge in the event of an outbreak [14–16]. To explore the pattern of aggregation outbreaks and to provide a guideline for decision-making on the future control of population aggregation activities, we collected the data of all aggregation outbreaks occurring in Jilin Province from January 12, 2020, to August 18, 2021. Based on the distribution of cases over time, the data collected were divided based on the three phases of the COVID-19 epidemic; the epidemiological characteristics of the cases were investigated to determine the occurrence of different types of aggregated cases, compare the changes in the characteristics of aggregated epidemics over time, identify the high-risk groups and places prone to the spread of aggregated epidemics, and provide

useful reference values for health departments to avoid future aggregated epidemics and identify effective prevention and control measures.

## Materials and methods

### Study area

Jilin is located in north-eastern China. Jilin Province has nine regions: Baicheng, Songyuan, Siping, Changchun, Liaoyuan, Jilin, Tonghua, Baishan, and Yanbian Korean Autonomous Prefecture [17]. According to the latest statistical yearbook report, the Jilin Province had a population of 27,000,040 at the end of 2018. The Jilin Province has a total area of $18.74 \times 10^4$ $km^2$ and is located in a typical north temperate continental monsoon climate area. The annual mean temperature is 5˚C–8.6˚C, and the average precipitation is between 350 and 1,000 mm.

### Data collection

As of August 18, 2021, information from 578 COVID-19 patients in Jilin, China were collected, including 552 with confirmed cases (1 died) and 26 with asymptomatic cases. The database included baseline information (sex, age, occupation, address, and area), case classification (asymptomatic infection and confirmed case), clinical severity (asymptomatic cases, mild, normal, severe, and critical cases), infection routes (imported cases, close contact with local cases, and close contact with provincial cases, except one that was undetermined), troubleshooting method (outpatient service and active screen), date of onset (except the 26 patients with asymptomatic infections), date of diagnosis, and date of discharge (except for one death). In addition, 74 patients with cluster events also included case type and number of close contacts. In the first phase of an aggregated outbreak, the type of patient's case, number of close contacts, and intergenerational relationships were investigated.

### Case definition

According to the definition of the Technical Guidelines for Epidemiological Investigation in the "New Coronavirus Pneumonia Prevention and Control Plan (Fourth Edition)" announced by the National Health Commission of the People's Republic of China [18], the clustered epidemic was defined as a transmission occurring in a small place (such as a household, a construction site, a unit, etc.) observed within 14 days; two or more patients have confirmed cases or asymptomatic infection, and a human-to-human transmission is possible due to close person-to-person contact or a joint exposure. Patients with confirmed cases should have one of the following two etiological factors: 1. The respiratory or blood specimens are consistently positive for new coronavirus nucleic acids on fluorescent reverse transcription polymerase chain reaction, and 2. sequencing of the respiratory or blood viral gene is highly homologous to the new coronavirus.

The beginning and end of the epidemic phase were defined artificially. When the first case is discovered in a place with no infectious source and no outbreak, it is considered the beginning of an epidemic phase. The end of this epidemic phase is considered when no new case occurs in Jilin Province more than 14 days after the end of the infectious period of the last case.

### Laboratory testing

Laboratory tests included nasal, throat, and anal swabs as well as blood and sputum tests.

## Global spatial autocorrelation

The global Moran's *I* coefficient was calculated using the OpenGeoDa 1.2.0 software to detect the overall spatial autocorrelation of the three phases of the epidemic in Jilin Province, with the Moran's coefficients ranging from −1 to 1. A value of > 0 indicates a positive spatial correlation; the larger the value, the stronger the spatial aggregation. A value of < 0 indicates a negative spatial correlation, while a value closer to −1 indicates greater spatial variability. A value of zero indicated the absence of spatial correlation. The significance of Moran's coefficient was also assessed based on the *Z* and *P* values.

## Local spatial autocorrelation

The OpenGeoDa 1.2.0 software was used to calculate the local spatial Moran's *I* coefficient (local indicators of spatial autocorrelation) and to analyze the local spatial autocorrelation in four cases: high-high, high-low, low-high, and low-low clusters. The ArcGIS 10.2 software was used to visualize and draw the aggregation maps.

## Statistical analysis

All data were entered into Excel 2019 and analyzed using the SPSS 21.0 (IBM Corp, Armonk, NY, USA) statistical package; was used to describe all measures, while the chi-square test was used to compare the count data. The significance level was set to α = 0.05.

# Results

## Epidemiologic description

According to the definition of epidemic stages, as of August 18, 2021, three phases of outbreaks occurred in Jilin Province: 97 cases from January 12, 2020 to February 19, 2020, 46 cases from April 25, 2020, to May 23, 2020, and 435 cases from January 10, 2021, to February 9, 2021. A total of 19 aggregated outbreaks occurred, with the first phase involving 17 aggregated outbreaks, and the second and third phases each involving one aggregated outbreak. The main modes of transmission of the three phases of epidemic are shown in Fig 1. A total of 23 non-clustered cases occurred in the first phase of the epidemic, with the largest number of cases (74 cases) involved in clustered events, accounting for 76.29% of the total cases. In the second and third phases of the epidemic, all cases were clustered. The largest number of young adults in the 25–50-year age group accounted for 56.70% and 58.70% of cases in the first and second phases of the epidemic, respectively; meanwhile, the largest number of older people in the 50–90-year age group accounted for 62.76% of the cases in the third phase of the epidemic (Fig 2). The first phase of the epidemic had the highest proportion of male patients (56.70%) compared with the second and third phases (43.48% and 44.83%, respectively). In terms of the occupational distribution of all three phases of the epidemic, the largest number of patients were domestic workers and unemployed persons, accounting for 20.62%, 28.26%, and 40.23%, respectively, whereas retired persons accounted for 24.60% of the cases in the third phase of the epidemic (Table 1). For each event, the time of onset of the first symptomatic case was determined (first symptomatic case ≠ initial case). As it was not possible to record the time of onset of asymptomatic infected persons, these cases were not described. The 19 clusters of patients with symptomatic infections at aggregated times that occurred in the three phases of the outbreak were analyzed, with the largest number of clustered events (six) occurring on January 22. The main clustering outbreaks commonly occurred between January 17, 2020, and February 14, 2020, with a total of 17 outbreaks (Fig 3). Fewer cluster outbreaks occurred in the later,

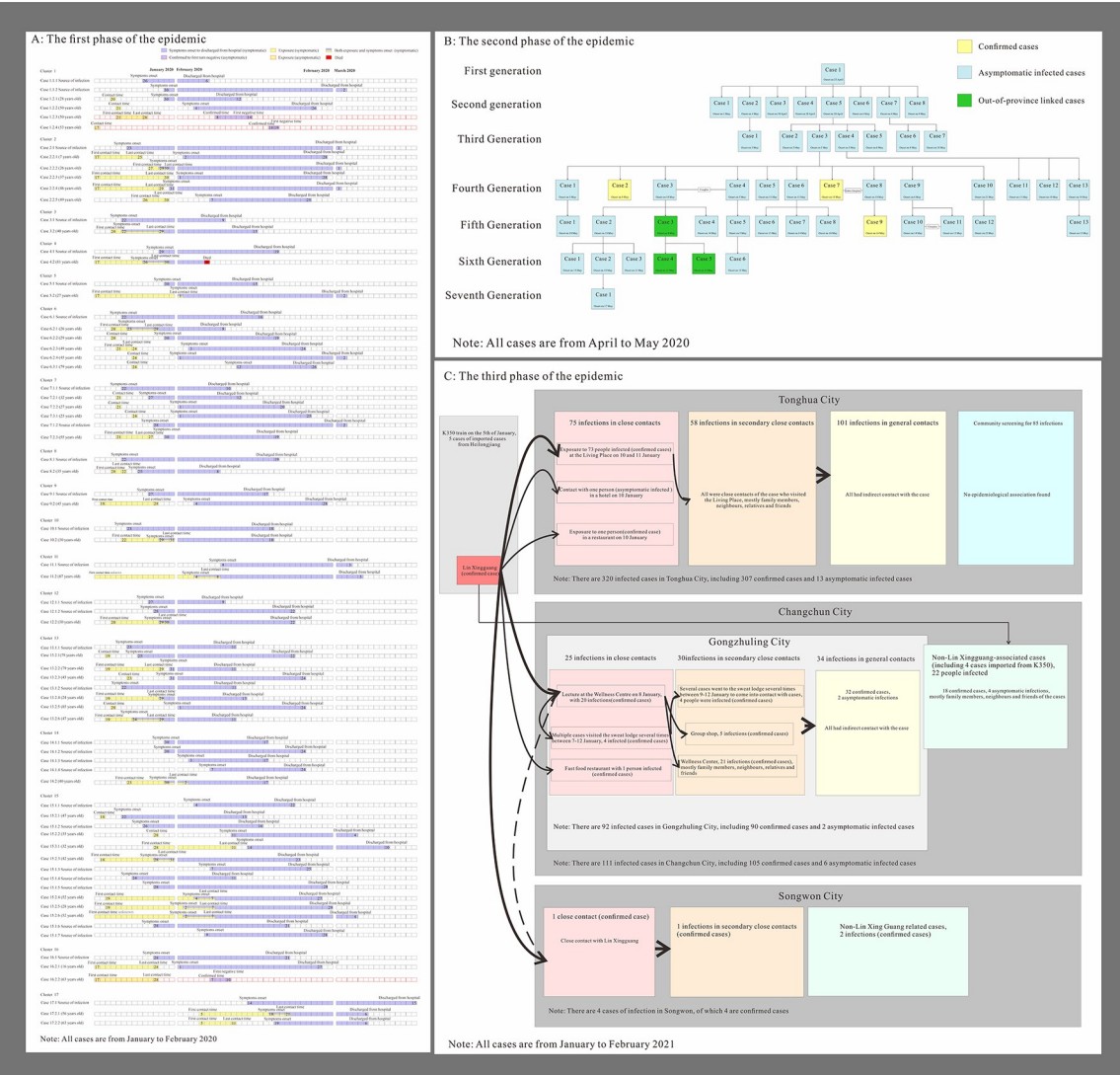

**Fig 1. Three phases of COVID-19 outbreaks transmission mode in Jilin Province.**

longer period, with two cluster outbreaks occurring on April 25, 2020, and January 10, 2021, respectively. The last two outbreaks involved more people compared with the previous 17 outbreaks, with the overall number of cluster events showing an initial decreasing trend followed by an increasing trend before finally decreasing. The largest number of cases occurred on January 10, 2021 (435 cases).

In the first phase of the outbreak, Changchun had the highest number of cases (22 patients), followed by Siping (10 patients). The second stage of the outbreak occurred mainly in Jilin and included 45 cases. The third phase of the outbreak occurred in Tonghua with 320 cases. The clustered events occurred in six regions of the Jilin Province, as shown in Fig 4. The two incidents with the highest number of cases involved in the aggregated events were located in Jilin (spreading to Changchun) and Tonghua (spreading to Changchun and Songyuan), with six aggregated events reported in Siping and five aggregated events in Changchun.

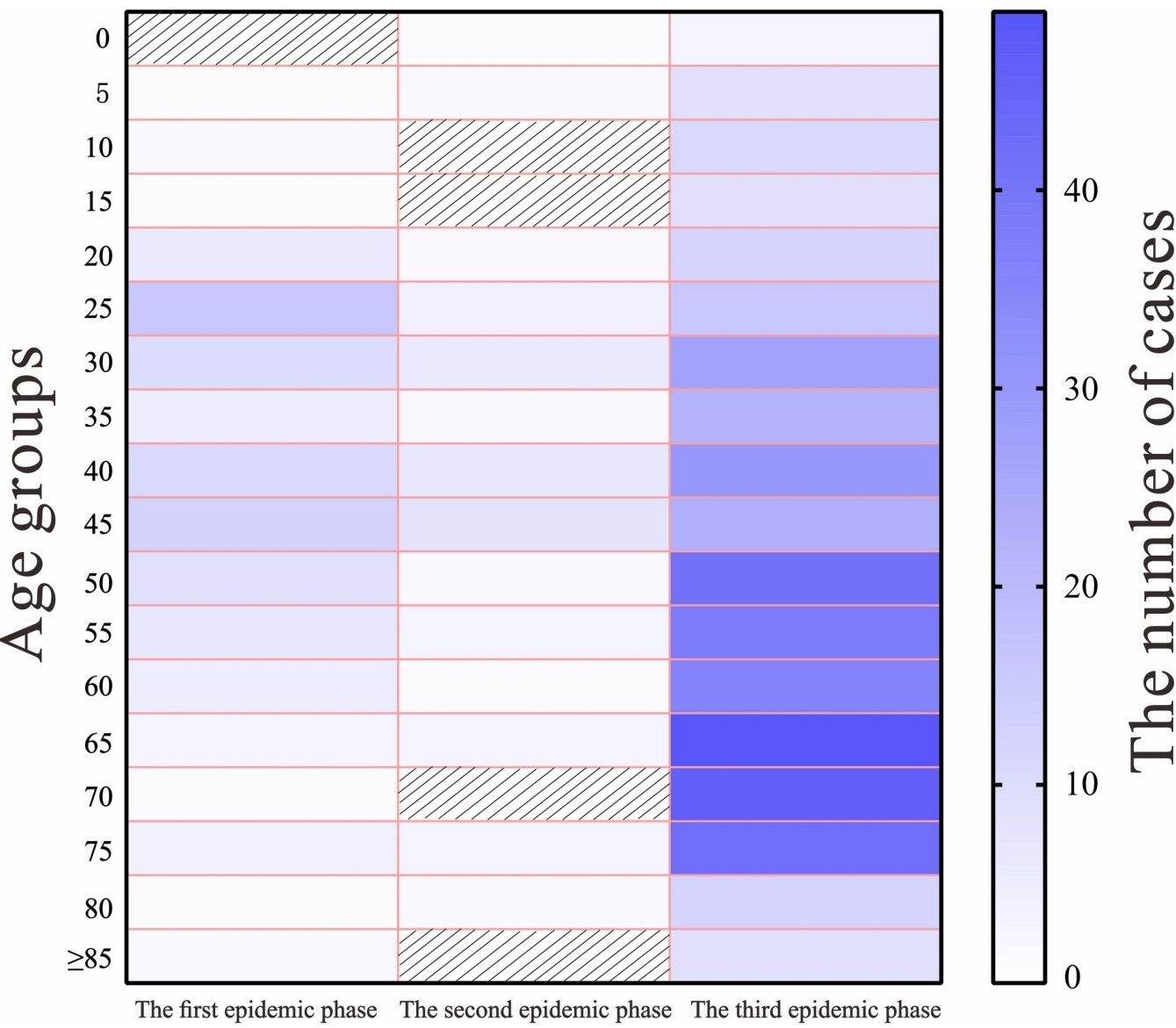

**Fig 2. Age distribution of the three phases of the COVID-19 outbreaks in Jilin Province.**

As the contact time of disseminated cases exposed to the associated cases was difficult to identify, only the incubation period of cases involved in aggregated outbreaks was calculated in this study. Based on the information obtained from the flow survey, the contact time and onset interval of renewed cases along with the source of infection were used to calculate the incubation period. The median incubation periods for the first, second, and third phases of the outbreak were 10 days, 8 days, and 5 days, respectively. There are four types of aggregated events in Jilin Province. Each type of event can involve multiple modes of aggregation, all of which include household contact, staying in public places, attending gatherings, and work; the various types of aggregated events are shown in Table 2. All the clustered events in Jilin Province involved household contact and five compound contact aggregated events.

**Table 1. Population characteristics of the three phases of outbreaks in Jilin Province.**

| Characteristics | The first phase of the epidemic | The second phase of the epidemic | The third phase of the epidemic | Test statistics | P value |
|---|---|---|---|---|---|
| **Gender** | | | | | |
| Male | 55 | 20 | 195 | $\chi^2 = 4.702$ | 0.094 |
| Female | 42 | 26 | 240 | | |
| **Age** | | | | | |
| 0- | 0 | 1 | 3 | F = 12.764 | < 0.05 |
| 5- | 1 | 2 | 9 | | |
| 10- | 2 | 0 | 11 | | |
| 15- | 1 | 0 | 9 | | |
| 20- | 6 | 2 | 12 | | |
| 25- | 16 | 4 | 16 | | |
| 30- | 10 | 6 | 27 | | |
| 35- | 5 | 2 | 22 | | |
| 40- | 11 | 7 | 30 | | |
| 45- | 13 | 8 | 23 | | |
| 50- | 9 | 2 | 41 | | |
| 55- | 7 | 3 | 38 | | |
| 60- | 5 | 1 | 36 | | |
| 65- | 3 | 3 | 49 | | |
| 70- | 1 | 0 | 46 | | |
| 75- | 4 | 3 | 42 | | |
| 80- | 1 | 2 | 12 | | |
| 85- | 2 | 0 | 9 | | |
| **Occupation** | | | | | |
| Staff and Officers | 16 | 9 | 5 | $\chi^2 = 148.660$ | < 0.05 |
| Workers | 6 | 1 | 9 | | |
| Domestic and non-working | 20 | 13 | 175 | | |
| Teachers | 0 | 1 | 3 | | |
| Retirees | 14 | 4 | 107 | | |
| Farmers | 10 | 8 | 10 | | |
| Diaspora children | 0 | 3 | 4 | | |
| Business Services | 8 | 5 | 13 | | |
| Students | 6 | 1 | 28 | | |
| Medical staff | 4 | 1 | 13 | | |
| Other | 9 | 0 | 15 | | |
| Not available | 4 | 0 | 53 | | |
| **Incubation period (median)** | 10 | 8 | 5 | F = 29.550 | < 0.05 |

## Statistical tests for clustered and non-clustered cases

The viral strains detected in the different phases of the epidemic and the prevention and control policies at that time vary considerably; therefore, the comparison of different types of cases in the same period is meaningful. As all cases in the second and third phases of the epidemic were aggregated epidemic-involved cases, all patients in the first phase were selected for this study and divided into aggregated and non-aggregated epidemic cases for comparison. Statistical methods were used to compare the age, occupation, urban and rural distribution, severity, mode of infection, and method of detection between the 74 patients involved in the 17 aggregated outbreaks and those involved in the non-aggregated outbreaks in the first phase of the

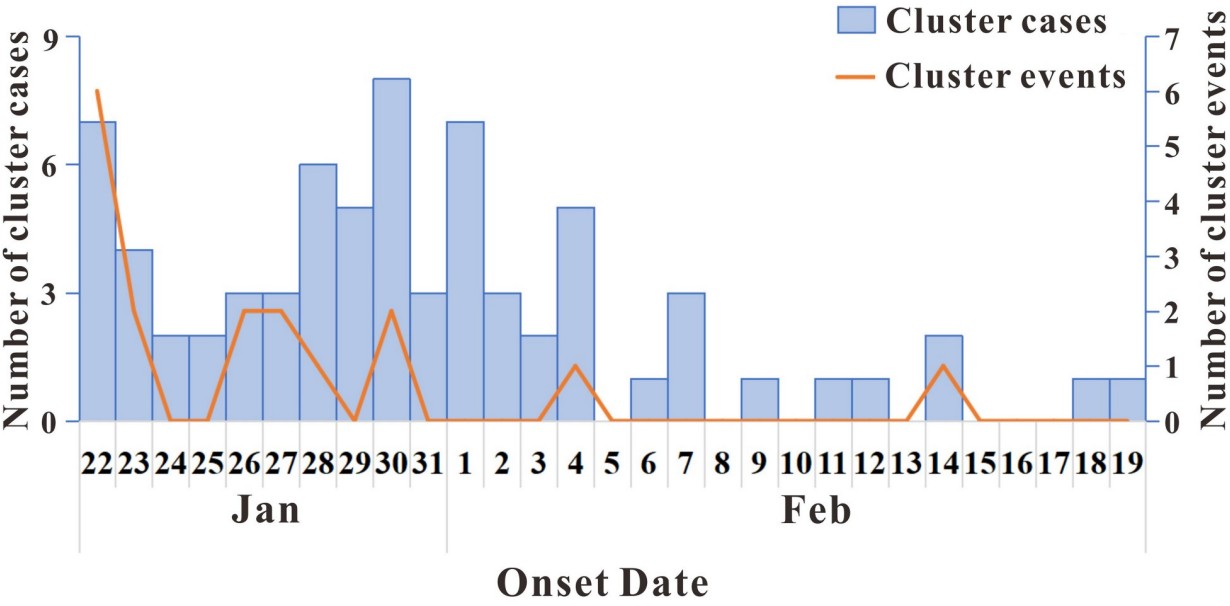

**Fig 3. Onset time of cluster events and cluster cases.**

outbreak. Results (Table 3) showed that the differences in detection methods between the clustered and non-clustered cases were statistically significant. Active screening was performed in 44 of the 74 cases and two of the 23 non-clustered cases; thus, the aggregated cases were more likely to be detected by active screening than by evaluation in outpatient clinics; this finding indicates that the aggregated cases in the first phase of the outbreak in Jilin Province were more likely to be detected by active screening compared with the non-clustered cases. In addition, the difference in the number of days from diagnosis to discharge between aggregated and non-aggregated cases was statistically significant. The median number of days from diagnosis to discharge was 5.5 days longer in the aggregated cases compared with that in the non-aggregated cases; this result indicated that the number of days from diagnosis to discharge was longer in the aggregated cases in the first phase of the outbreak in Jilin Province compared with that in the non-aggregated cases. The factors other than the test method and time from diagnosis to discharge did not differ significantly between the aggregated and non-aggregated cases.

## Statistical tests for differences between source of infection and sequel cases in clustered cases

The local health authorities immediately implemented measures such as city lockdown and travel restrictions at the beginning of the second and third phases of the outbreak; all cases in these two phases were local secondary cases, except one, which was an imported case. Therefore, all 17 cases from the first phase of the epidemic were selected for this study, and the differences between the infectious and sequelae cases in the same time period were compared. Statistical tests were conducted to examine the differences in the distribution of infectious and sequelae cases by sex, occupation, urban and rural distribution, severity, mode of infection, method of detection, time of onset to time of diagnosis, time of diagnosis to time of discharge, and time of onset to time of discharge. Based on the results of the tests, significant differences were observed in the in the mode of infection and method of detection between infectious and sequela cases in the first phase of clustered events (Table 4). Imported cases accounted for 24

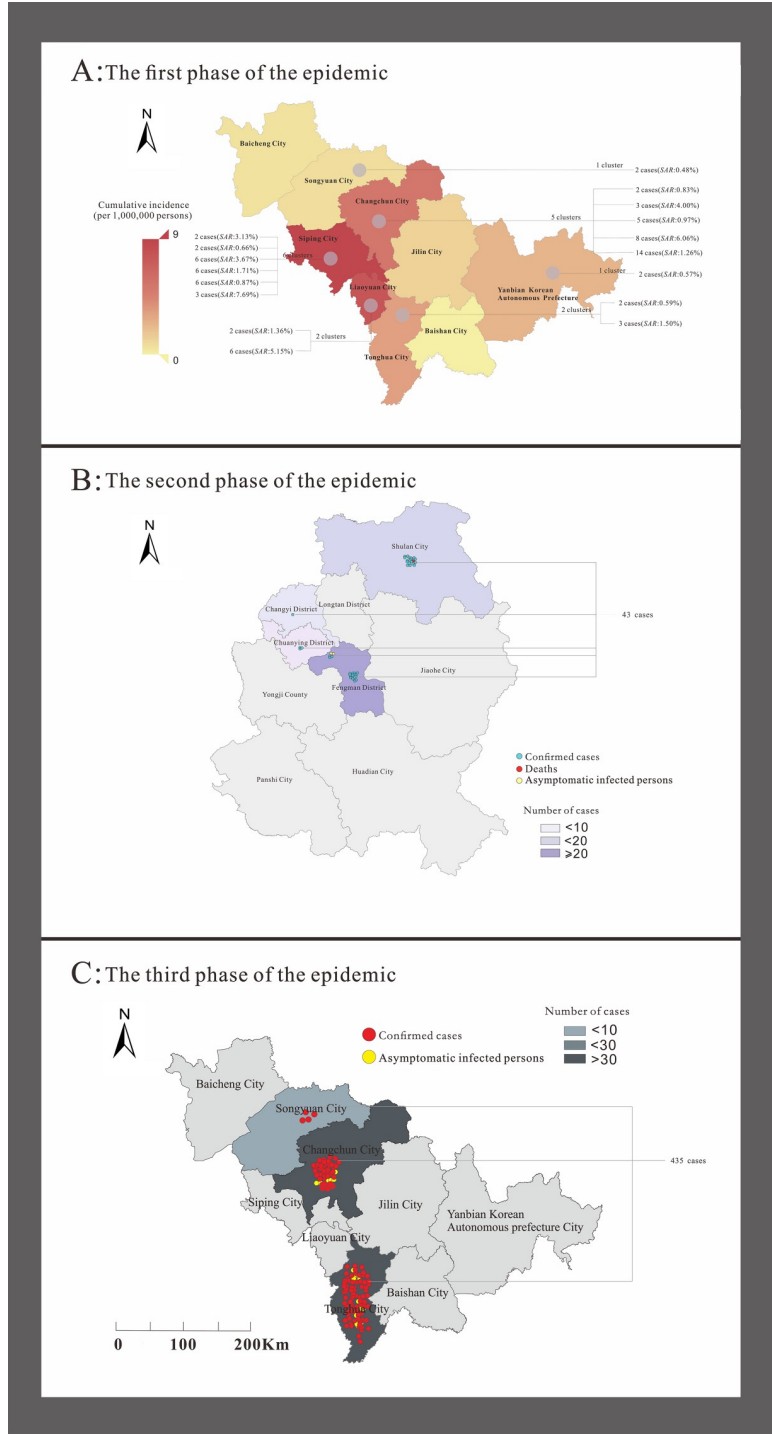

**Fig 4. Regional distribution of three phases of COVID-19 outbreaks in Jilin Province.** The map depicted in this figure was taken from Datamap (http://datamaps.github.io/).

of the 30 source cases and two of the 44 sequel cases; this finding indicates that there were more imported cases among the sources of infection than among the sequel cases. Cases detected in outpatient clinics accounted for 19 of the 30 sources of infection and four of the 44

**Table 2. Types of cluster events.**

| Cluster types | Event | Cases | Composition ratio (%) | Total case composition ratio (%) |
|---|---|---|---|---|
| Family | 14 | 51 | 73.69 | 9.19 |
| Family & Public place | 1 | 3 | 5.26 | 0.54 |
| Family & Public place & Gathering | 1 | 6 | 5.26 | 1.08 |
| Family & Public place & Gathering & Work | 3 | 495 | 15.79 | 89.19 |
| Total | 19 | 555 | 100.00 | 100.00 |

sequel cases, indicating that the source of infection was more likely to be detected in outpatient clinics than the sequel cases. The differences in the other factors between the two populations were not significant.

## Statistical tests for the differences between the three phases of the epidemic

To investigate the differences in case information between the different phases of the epidemic, all cases were divided into three phases according to the period in which the cases were collected, and the differences in the baseline information (gender, age, and occupation) and incubation period between the three phases of the epidemic were examined. A significant difference was observed in age, occupation, and incubation period, except for sex, between the three phases of the epidemic (Table 1). A two-by-two comparison of the three phases of the epidemic in terms of age and incubation period was conducted. A significant difference was observed in the proportion of different age groups between the third and first two phases of the epidemic, while no difference was observed in the age groups between the first and second phases. The third phase of the epidemic had the highest proportion of older people aged 50–90 years (62.76%); this result suggests that the third phase of the epidemic had a higher proportion of middle-aged and older people than the first two phases. Significant differences were found in the percentage of incubation periods between the third and first two phases, while no differences were found in the incubation periods between the first and second phases. The mean incubation periods in all three phases were 10 days, 8 days, and 5 days, respectively, indicating that the incubation period of the third phase was shorter than those of the first two phases.

## Spatiotemporal analysis

Moran's I coefficients did not differ significantly between the two epidemic phases ($P>0.05$), except for the second phase of epidemics in Jilin ($P < 0.05$). The second phase of the epidemic showed a significant spatial variability (Moran's $I < 0$, $P < 0.05$) in Jilin City (Table 5). For global spatial autocorrelation testing, we performed a local spatial autocorrelation analysis of each district and city in Jilin in the second phase of the epidemic and observed an uneven regional distribution of COVID-19 incidence in Jilin City. The high-low cluster area is concentrated in Shulan City. The results are shown in Fig 5.

## Discussion

COVID-19 has posed great challenges to all health professionals worldwide. China has experienced a large outbreak, and its economy is gradually stabilizing; however, the cumulative number of overseas imported infections is slowly increasing. Except in Wuhan, China, the initial cases were mainly imported cases, and the number of infections eventually increased due to the occurrence of cluster events. During the Spring Festival, China adopted measures such as sealing off the city, prohibiting the entry of foreign populations, closing public places, and extending holidays to reduce the incidence of imported and cluster cases, which achieved good results

**Table 3. Characteristics analysis of the first phase of the epidemic in Jilin Province.**

| Characteristic | Non-Cluster cases (n = 23) | Cluster cases (n = 74) | Test statistics | P value |
|---|---|---|---|---|
| **Gender** | | | | |
| Male (n = 55) | 14 | 41 | $\chi^2 = 0.213$ | 0.810 |
| Female (n = 42) | 9 | 33 | | |
| **Occupation** | | | | |
| Food and beverage industry (n = 1) | 0 | 1 | - | - |
| Officers (n = 16) | 2 | 14 | | |
| Workers (n = 6) | 3 | 3 | | |
| Public places attendant (n = 2) | 0 | 2 | | |
| House-workers and unemployed (n = 20) | 7 | 13 | | |
| Retirees (n = 14) | 3 | 11 | | |
| workforce (n = 1) | 0 | 1 | | |
| Farmers (n = 9) | 0 | 9 | | |
| Others (n = 6) | 0 | 6 | | |
| Business services (n = 8) | 3 | 5 | | |
| Students (n = 6) | 3 | 3 | | |
| Medical staff (n = 4) | 1 | 3 | | |
| Unknown (n = 4) | 1 | 3 | | |
| **Area** | | | | |
| Urban (n = 83) | 21 | 62 | - | 0.508* |
| Rural (n = 14) | 2 | 12 | | |
| **severity** | | | | |
| Asymptomatic cases (n = 4) | 1 | 3 | Z = -0.202 | 0.840 |
| Mild Cases (n = 39) | 10 | 29 | | |
| Normal Cases (n = 48) | 10 | 38 | | |
| Severe Cases (n = 5) | 2 | 3 | | |
| Critical Cases (n = 1) | 0 | 1 | | |
| **Age (n)** | | | | |
| 0–9 (1) | 0 | 1 | - | - |
| 10–19 (3) | 1 | 2 | | |
| 20–29 (22) | 8 | 14 | | |
| 30–39 (15) | 5 | 10 | | |
| 40–49 (24) | 2 | 22 | | |
| 50–59 (16) | 5 | 11 | | |
| 60–69 (8) | 1 | 7 | | |
| 70–79 (5) | 1 | 4 | | |
| 80- (3) | 0 | 3 | | |
| Median (Range) | 33 (11,70) | 44.5 (7,87) | | |
| **Infection ways** | | | | |
| Imported cases (n = 45) | 19 | 26 | - | - |
| Close contact with local cases (n = 8) | 0 | 8 | | |
| Close contact with provincial cases (n = 43) | 4 | 39 | | |
| Unknown (n = 1) | 0 | 1 | | |
| **The detected method** | | | | |
| Outpatient found (n = 51) | 21 | 30 | $\chi^2 = 18.135$ | 0.000 |
| Active screening (n = 46) | 2 | 44 | | |
| **Case classification** | | | | |

*(Continued)*

**Table 3.** (Continued)

| Characteristic | Non-Cluster cases (n = 23) | Cluster cases (n = 74) | Test statistics | P value |
|---|---|---|---|---|
| Confirmed cases (n = 93) | 22 | 71 | - | 1.000* |
| Asymptomatic cases (n = 4) | 1 | 3 | | |
| **Days from illness onset to diagnosis** | | | | |
| Median (Range) | 7 (2,14) | 5 (0,13) | Z = 1.762 | 0.78 |
| **Days from diagnosis to discharged from hospital time** | | | | |
| Median (Range) | 11.5 (5,27) | 17 (8,28) | Z = -1.973 | 0.048 |
| **Days from onset time to discharged from hospital time** | | | | |
| Median (Range) | 19.5 (13,31) | 22 (11,38) | t = 0.792 | 0.431 |

* = Fisher's exact probability test.

[19]. In Beijing, Shanghai, and other provinces and cities in China, the number of cluster cases accounted for the number of confirmed cases (50%–80%) [20,21]. The aggregated cases in Jilin Province suggested that most of the aggregated outbreaks involved family aggregation.

The COVID-19 outbreak in Jilin Province can be divided into three phases according to temporal distribution. The first phase of the outbreak involved 17 aggregated outbreaks, the second phase involved one aggregated outbreak, and the third phase involved one aggregated outbreak. The median incubation periods of the cases in all three phases of the outbreak in Jilin Province were 10 days, 8 days, and 5 days, respectively, with the longest of all cases being 19 days and the shortest being 0 days; that is, the symptoms appeared on the day of exposure. Based on previous studies, the longest incubation period for COVID-19 is 19 days [22]. As most of the first cases were imported, data on their exposure time were difficult to obtain; therefore, the incubation period of the renewed cases is more representative of the actual situation of COVID-19 in Jilin Province. Previous studies have indicated that the average incubation period of COVID-19 is 5 days [7]. The differences between the present study and previous studies may be due to the different strains of the transmitted viruses across regions. Results of statistical tests showed that the incubation period of the third stage of the epidemic differs from those of the first two stages, presumably due to the differences in the source of virus, thus leading to the variations in the incubation period of the infected cases [23].

The third stage of the outbreak was discovered through a flow survey, in which a lecturer who had already been infected with COVID-19 conducted a health promotion class and caused mass transmission of COVID-19 to the audience. Since the outbreak occurred during the winter, the lecture room was not properly ventilated as it was extremely cold outside; hence, most of the people who attended the lecture were infected. In addition, because the topic of the lecture was health related, most of the older adults were interested to listen as they were overly concerned about their health and had sufficient time to attend the event; hence, most of the attendees were unemployed or retired, which explains the difference in age and occupation between the third and first two stages of the epidemic.

This study found that most cases had the ability to transmit to the next generation of cases 2–7 days prior to the onset of symptoms; that is, they were infectious during the incubation period, which was similar to the findings of other studies related to aggregated outbreaks [10,24]. The epidemiological survey data of patients showed that most people during the aggregated outbreak had not been exposed to the infected area and had no contact with symptomatic patients, yet they still had the possibility of being infected; this finding confirmed the hypothesis that some asymptomatic patients also had the ability to spread the infection. Thus,

**Table 4. Characteristics of cluster cases.**

| Characteristic | Source of infection(n = 30) | Secondary cases (n = 44) | Test statistics | P value |
|---|---|---|---|---|
| **Gender** | | | | |
| Male (n = 41) | 20 | 21 | $\chi^2$ = 2.59 | 0.153 |
| Female (n = 33) | 10 | 23 | | |
| **Occupation** | | | | |
| Food and beverage industry (n = 1) | 1 | 0 | - | - |
| Officers (n = 14) | 8 | 6 | | |
| Workers (n = 3) | 1 | 2 | | |
| Public places attendant (n = 2) | 0 | 2 | | |
| Houseworkers and unemployed (n = 13) | 3 | 10 | | |
| Retirees (n = 11) | 5 | 6 | | |
| workforce (n = 1) | 1 | 0 | | |
| Farmers (n = 9) | 1 | 8 | | |
| Others (n = 6) | 4 | 2 | | |
| Business services (n = 5) | 3 | 2 | | |
| Students (n = 3) | 1 | 2 | | |
| Medical staff (n = 3) | 1 | 2 | | |
| Unknown (n = 3) | 1 | 2 | | |
| **Area** | | | | |
| Urban (n = 62) | 27 | 35 | - | 0.339* |
| Rural (n = 12) | 3 | 9 | | |
| **severity** | | | | |
| Asymptomatic cases (n = 3) | 0 | 3 | - | 4.92* |
| Mild Cases (n = 29) | 13 | 16 | | |
| Normal Cases (n = 38) | 16 | 22 | | |
| Severe Cases (n = 3) | 0 | 3 | | |
| Critical Cases (n = 1) | 1 | 0 | | |
| **Cases type** | | | | |
| Imported cases (n = 45) | 24 | 2 | $\chi^2$ = 44.562 | 0 |
| Non-imported cases (n = 45) | 6 | 42 | | |
| **The detected method** | | | | |
| Outpatient found (n = 23) | 19 | 4 | $\chi^2$ = 24.501 | 0 |
| Active screening (n = 51) | 11 | 40 | | |
| **Age (n)** | | | | |
| Median (Range) | 44.5 (10,77) | 44 (7,89) | t = 0.045 | 0.965 |
| **Days from illness onset to diagnosis** | | | | |
| Median (Range) | 5 (0,11) | 6 (0,13) | Z = -0.181 | 0.856 |
| **Days from diagnosis to discharged from hospital time** | | | | |
| Median (Range) | 17 (8,28) | 18 (8,25) | Z = -0.48 | 0.962 |
| **Days from onset time to discharged from hospital time** | | | | |
| Median (Range) | 21 (11,38) | 22.5 (11,32) | Z = -0.137 | 0.891 |

* = Fisher's exact probability test.

the neglect of the detection and management of asymptomatic infected persons in the early stages of the COVID-19 outbreak resulted in the widespread transmission. At the first sign of the epidemic, the close contacts of the infected population should be identified, and the management and isolation of the closely connected and sub-closely connected populations should

**Table 5. Global autocorrelation of the incidence of three phases of COVID-19 outbreaks in Jilin Province.**

| The three phases of the epidemic in Jilin Province | Moron's I | Z value | P value |
|---|---|---|---|
| The first phase | 0.049 | 0.754 | 0.221 |
| The second phase | -0.301 | -1.31 | 0.045 |
| The third phase | -0.119 | 0.020 | 0.402 |

be strengthened to effectively limit the spread of the epidemic. The detection methods used between the first and subsequent cases differed. The predominance of active screening as a detection method for sequel cases suggests that active screening can detect patients who may still be in the incubation period and is an effective method of controlling the spread of the disease. Aggregate cases had a higher rate of active screening than non-aggregate cases. Among the aggregated cases, the proportion of active screening was higher among renewed cases than among first cases, indicating that the local health authorities monitored and managed the close contacts in a timely manner. Furthermore, most of the renewed cases were detected through active testing, effectively controlling latent cases and avoiding the spread of the virus in mostly unknown situations. At the same time, significant differences were observed in the length of hospital stay between the aggregated cases and non-aggregated cases. All patients were discharged with a negative nucleic acid test over a period of time, and the aggregated cases were hospitalized for a longer period of time, which indicates that the aggregated outbreaks involved

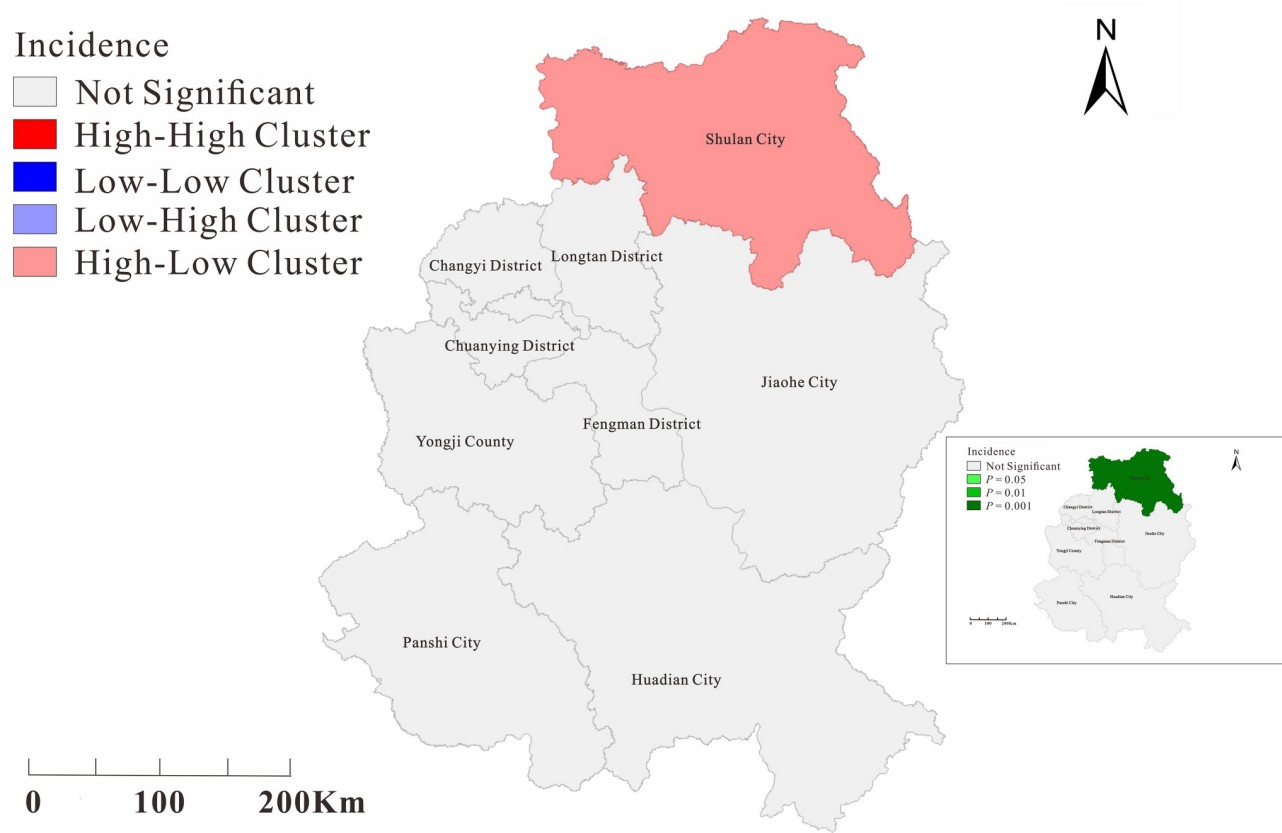

**Fig 5. Local spatial autocorrelation analysis of the second phase of COVID-19 incidence in Jilin Province.** The map depicted in this figure was taken from Datamap (http://datamaps.github.io/).

earlier admissions. This laterally proves that the local health department isolated their close contacts in a timely manner, allowing the source of infection to be detected early through active screening. The largest proportion of imported cases was detected in the first phase of the cluster outbreak, and the population size per cluster outbreak was smaller than that in the last two phases of the outbreak. This is due to the fact that during the first phase of the COVID-19 outbreak in Jilin Province, there was an inflow of a large number of cases from outside the province flowed into Jilin Province due to uncertainty and inexperience in the mode of transmission of the disease, and the failure to adopt contact avoidance methods such as city closures and travel restrictions in the first hours of the epidemic. However, when the disease was confirmed to be contagious, the relevant health authorities immediately implemented mandatory quarantine and city closure measures; hence, each cluster of outbreaks only involved a relatively small number of people. The cases in the latter two phases of the outbreak were all related to each other and were found in close and sub-close contacts of the same imported case, suggesting that sealing the city was effective in avoiding the inflow of cases from outside the province. The outbreaks in both phases were found to be caused by gatherings in public places and therefore involved a large number of people; although the outbreaks were effectively controlled at this stage, the focus of prevention and control should shift to the restriction of large gatherings in public places. For imported cases where various conditions permit (e.g., time and money), if you live with your relatives, home isolation is not recommended, or the family members need to take protective measures when they are at home. Infections during gatherings and common exposures occur mainly in places with a high population density and within a short period of time. Therefore, effective measures, such as controlling the flow of people, should be implemented in densely populated areas. Therefore, it is necessary to abolish any type of in-person gatherings [25]. For example, the conduct of online classes at home significantly reduced the spread of COVID-19. However, as the mobility of people returning to work and classes increases, this also increases the possibility of spreading the infection. The second and third phases of the outbreak in Jilin Province were caused by gatherings in public places under strict control of imported cases; therefore, it is even more important to avoid gatherings in the current epidemic situation to avoid the possibility of outbreaks caused by mass gatherings.

No spatial correlation was found between the differences in COVID-19 incidence rates across urban areas for the other two phases of the epidemic, except for the second phase; this result implies that during the first and third phases of the COVID-19 outbreak, cases showed a random distribution across municipalities and were not regionally clustered, which may be due to the fact that the epidemic had been spreading across municipalities in Jilin Province for several days by the time cases were detected. By contrast, the second phase of the epidemic in Jilin City, Jilin Province, was detected early at the beginning, and prevention and control measures were implemented quickly; thus, few cases spilled over from Shulan City, showing high-low aggregation in Shulan City and implying the risk of spreading the epidemic from Shulan City to the surrounding municipalities. The results of this spatiotemporal analysis are consistent with the epidemiological distribution of this study.

The proportion of imported cases in the first phase of the cluster epidemic was relatively high, while the size of the population per cluster epidemic was smaller than that in the last two phases of the epidemic, which was due to the fact that during the first phase of the COVID-19 epidemic in Jilin Province, there was an inflow of a large number of cases from outside the province into the Jilin Province due to uncertainty and inexperience with the mode of transmission of the disease and the failure to adopt contact avoidance methods such as city closures and travel restrictions in the first hours of the epidemic. However, after confirming that the disease was contagious, the relevant health authorities immediately implemented mandatory

quarantine and city closure measures; hence, each cluster of outbreaks only involved a relatively small number of people. The cases in the second and third phases of the outbreak were linked to each other, with cases found in close and sub-close contacts of the same imported case, indicating that the closure of the city was effective in avoiding the inflow of cases from outside the province. The outbreak of the second and third phases of the epidemic was caused by gatherings in public places, thus involving a large number of people; although the outbreak was effectively controlled at this stage, the focus of prevention and control should shift to the avoidance of large gatherings in public places.

Infections and general exposure at gatherings occur mainly in areas with a high population density. Therefore, effective measures, such as crowd control, should still be implemented in densely populated areas. Gatherings must be eliminated [25]. For example, the conduct of online class at home significantly reduced the spread of COVID-19. However, with the increased movement to and from home and tourist travel during holidays, this made it possible for the outbreak to spread. The second and third phases of the epidemic in Jilin Province were caused by gatherings in public places under the strict control of imported cases. Therefore, in the current epidemic situation, gatherings should be restricted to avoid the occurrence of an epidemic caused by mass gatherings.

## Limitation

This study had some limitations. First, the total number of cases involved in the three phases of the outbreak varied considerably and therefore may be biased when performing statistical analyses. Some of the aggregated events involved fewer cases, such as intra-household transmission that only involved two or three people, and therefore are not as convincing compared with the larger aggregated events. Due to the large number of people involved in the third phase of the epidemic, no clear intergenerational relationships (e.g., intergenerational spacing and renewal rates) were presented in the flow survey data, making it difficult to analyze. The number of close contacts involved in this study has been located to the greatest extent possible, but there is a possibility of omission. Second, future studies should measure the spatial stratified heterogeneity (SSH) to further investigate the interregional transmission patterns of these three phases of the epidemic [26,27]. In this study, the lack of detailed case locations due to data quality limitations can lead to problems of spatial applicability when assessing the epidemiological spatial distribution. Second, this study aimed to investigate the methodology of aggregated outbreaks to inform the prevention and control of aggregated outbreaks, which is why SSH was not included. In future studies, we will attempt to measure the SSH and further elucidate the pattern of transmission of COVID-19 among different regions.

## Conclusion

Cluster cases comprised the highest component of the total number of cases. Surveillance of outbreaks is of utmost importance. In addition, family gatherings and high traffic areas should be avoided. Simultaneously, as the number of people returning to work and school increases, precautions should be taken to avoid the possibility of secondary outbreaks.

## Supporting information

**S1 Table. The number of reported cases in three waves of the COVID-19 outbreak in Jilin Province, China.**
(PDF)

## Acknowledgments

The authors would like to express their sincere gratitude to the following people, without whom the study would not have been possible: (1) study participants for providing data and (2) field investigators for collecting the data.

## Author Contributions

**Formal analysis:** Yifei Zhao.

**Methodology:** Rui Tian.

**Software:** Bonan Cao.

**Supervision:** Yang Zhang.

**Visualization:** Xi Sheng.

**Writing – original draft:** Qinglong Zhao, Meina Li, Laishun Yao.

**Writing – review & editing:** Yan Yu.

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
