## [Decision Letter · Decision Letter 0]

30 Jun 2022

PONE-D-21-36962Epidemiological clustered characteristics of COVID-19 in three phases transmission in Jilin Province, ChinaPLOS ONE

Dear Dr. Yan,

Thank you for submitting your manuscript to PLOS ONE. After careful consideration, we feel that it has merit but does not fully meet PLOS ONE’s publication criteria as it currently stands. Therefore, we invite you to submit a revised version of the manuscript that addresses the points raised during the review process. Please revise.

We look forward to receiving your revised manuscript.

Kind regards,

Academic Editor

PLOS ONE

**Journal requirements:**

3. PLOS requires an ORCID iD for the corresponding author in Editorial Manager on papers submitted after December 6th, 2016. Please ensure that you have an ORCID iD and that it is validated in Editorial Manager. To do this, go to ‘Update my Information’ (in the upper left-hand corner of the main menu), and click on the Fetch/Validate link next to the ORCID field. This will take you to the ORCID site and allow you to create a new iD or authenticate a pre-existing iD in Editorial Manager. Please see the following video for instructions on linking an ORCID iD to your Editorial Manager account: https://www.youtube.com/watch?v=_xcclfuvtxQ.

4. We note that [Figure 4] in your submission contain [map/satellite] images which may be copyrighted. All PLOS content is published under the Creative Commons Attribution License (CC BY 4.0), which means that the manuscript, images, and Supporting Information files will be freely available online, and any third party is permitted to access, download, copy, distribute, and use these materials in any way, even commercially, with proper attribution. For these reasons, we cannot publish previously copyrighted

maps or satellite images created using proprietary data, such as Google software (Google Maps, Street View, and Earth). For more information, see our copyright guidelines: http://journals.plos.org/plosone/s/licenses-and-copyright.

a. You may seek permission from the original copyright holder of Figure 4 to publish the content specifically under the CC BY 4.0 license. 

Reviewers' comments:

Reviewer's Responses to Questions

**Comments to the Author**

1. Is the manuscript technically sound, and do the data support the conclusions?

Reviewer #1: Partly

Reviewer #2: Yes

2. Has the statistical analysis been performed appropriately and rigorously? 

Reviewer #1: No

Reviewer #2: Yes

3. Have the authors made all data underlying the findings in their manuscript fully available?

Reviewer #1: No

Reviewer #2: Yes

4. Is the manuscript presented in an intelligible fashion and written in standard English?

Reviewer #1: No

Reviewer #2: No

5. Review Comments to the Author

Reviewer #1: 1. Provide mathematical strict definitions of the “waves” and “phases”;

2. Measure and attribute spatial stratified heterogeneity and interpret the findings in epidemiology;

3. The paper is not well structed and described, should be improved;

4. English should be edited by English native speakers.

Reviewer #2: The authors have made a serious attempt at analysing the three waves of transmission in a small province in China. The data is presented neatly.

However the conclusions do not fully reflect the results obtained.

English grammar, formation of sentences, spelling errors and syntaxal errors abound throughout the manuscript and need very urgent attention

6. PLOS authors have the option to publish the peer review history of their article (what does this mean?). If published, this will include your full peer review and any attached files.

Reviewer #1: No

Reviewer #2: No

---

## [Author Response · Author response to Decision Letter 0]

4 Nov 2022

We thank the reviewers and you for the opportunity to revise and resubmit our work as well as for their comments that helped us considerably improve the content and presentation of the paper. We have performed a very thorough and major revision of the paper, resulting in a significantly improved manuscript. We have also addressed reviewers’ comments line by line. The major changes in the manuscript appear in red.

We have provided our explanations and responses after the comments below.

Sincerely,

Tianmu Chen, Ph.D

State Key Laboratory of Molecular Vaccinology and Molecular Diagnostics, School of Public Health, Xiamen University

4221-117 South Xiang’an Road, Xiang’an District, Xiamen, Fujian Province, People’s Republic of China

Tel: +86-13661934715

Email: 13698665@qq.com

1.Please ensure that your manuscript meets PLOS ONE's style requirements, including those for file naming. 

Author response: Dear Editors, I ensure that my manuscript meets PLOS ONE's style requirements, including those for file naming. 

2.We note that you have indicated that data from this study are available upon request. PLOS only allows data to be available upon request if there are legal or ethical restrictions on sharing data publicly.

Author response: Dear Editors, the datasets used and analyzed during this current study are available from Dr. Qinglong Zhao (jlcdczql@126.com) upon reasonable request.

3.PLOS requires an ORCID iD for the corresponding author in Editorial Manager on papers submitted after December 6th, 2016. Please ensure that you have an ORCID iD and that it is validated in Editorial Manager. To do this, go to ‘Update my Information’ (in the upper left-hand corner of the main menu), and click on the Fetch/Validate link next to the ORCID field. This will take you to the ORCID site and allow you to create a new iD or authenticate a pre-existing iD in Editorial Manager. 

Author response: Dear Editors, We have a clear ORCID iD in POLS. 

4.We note that [Figure 4] in your submission contain [map/satellite] images which may be copyrighted. All PLOS content is published under the Creative Commons Attribution License (CC BY 4.0), which means that the manuscript, images, and Supporting Information files will be freely available online, and any third party is permitted to access, download, copy, distribute, and use these materials in any way, even commercially, with proper attribution. For these reasons, we cannot publish previously copyrighted maps or satellite images created using proprietary data, such as Google software (Google Maps, Street View, and Earth). For more information, see our copyright guidelines: http://journals.plos.org/plosone/s/licenses-and-copyright.

Author response: Dear Editors, We have contacted the original copyright holder with the Content Permission Form and upload the completed Content Permission Form or other proof of granted permissions as an ""Other"" file with my submission.

Reviewer #1：

1. Provide mathematical strict definitions of the “waves” and “phases”

Author response: Dear Reviewer, Thank you very much for your valuable suggestions. As this study analyses the three epidemic phases in Jilin Province in chronological order, the word "waves" has been changed to "phases" in all the text and images of this study to make the definition of the three epidemics in the article more precise. The definition of "phases" has also been added to the article, namely the beginning and end of the epidemic phase is defined artificially. When the first case appears in a place in the state of no infectious source and no case, it is considered the beginning of an epidemic phase, and the end of this epidemic phase is considered when no new case appears in Jilin Province more than 14 days after the end of the infectious period of the last case. See lines 200-203 for details.

2. Measure and attribute spatial stratified heterogeneity and interpret the findings in epidemiology

Author response: Dear Reviewer, thank you again for your suggestion for this study, which has been very useful for this study. We have very carefully reviewed the literature on the use of spatial stratified heterogeneity (SSH) in infectious diseases. The SSH method you mentioned and reported in the literature [1][2][3] should have been used in this study to produce more detailed results on this outcome. We made some attempts with relevant data and software. However, as this study only obtained reported data within nine delineated regions in Jilin Province, and not more detailed location data for individual cases, the crude data partitioning resulted in too few regions involved in the assessment to explore the results of its SSH. In our next study we will look for more detailed regional segmentation data or a larger spatial distribution study so that the method can be combined in a study of spatially stratified heterogeneous transmission across regions. Also, in the limitations of this paper we mention that this study should also measure SSH to better explain the inter-regional transmission patterns of these three outbreaks of aggregated outbreaks, but the lack of detailed case locations to assess epidemiological spatial distribution issues due to data quality limitations may lead to some spatial applicability issues with the results of the 3 phases of this study. However, as the aim of this study is to provide a method to control and prevent the spread of COVID-19 aggregated outbreaks, it will not have a large impact on the outcome of this study. See lines 513 and 520 for details.

[1] Hu B, Ning P, Qiu J, Tao V, Devlin AT, Chen H et al: Modeling the complete spatiotemporal spread of the COVID-19 epidemic in mainland China. Int J Infect Dis 2021, 110:247-257.doi: 10.1016/j.ijid.2021.04.021.

[2] Buttle J.M, Allen D.M, Caissie D, Davison B, Hayashi M, Peters D.L. Flood processes in Canada: regional and special aspects. Can Water Resour J Rev Can Ressour Hydr. 2016;41:7–30. doi: 10.1080/07011784.2015.1131629.

[3] Wang J.F, Zhang T.L, Fu B.J. A measure of spatial stratified heterogeneity. Ecol Indic. 2016;67:250–256. doi: 10.1016/j.ecolind.2016.02.052.

However, we made a spatially autocorrelated attempt to understand the spatial and temporal distribution of these three stages of COVID-19 in the aggregated epidemic in Jilin Province, which has explained the phenomenon in its epidemiology. We have made significant revisions. In the methods section, we have added methods for global and local spatial autocorrelation of the incidence of the three phases of the aggregated epidemic in Jilin Province; in the results section, we have added analysis of the spatial heterogeneity of regions in the three phases of the epidemic in Jilin Province. Finally, we have deepened the results section and revamped the discussion section. See lines 206 and 218, 347 and 354, and 466 and 476 for details.

3. The paper is not well structed and described, should be improved

Author response: Dear Reviewers, thank you again for your valuable suggestions for this study. Based on your suggestions, we have made extensive and detailed changes to the structure of this paper and have improved the logic and coherence of the article.

4. English should be edited by English native speakers.

Author response: Dear Reviewer,your valuable comments have been very helpful and we have sought help from the English editor at Editage (www.editage.cn) and have made significant language changes to this paper.

Reviewer #2: The authors have made a serious attempt at analysing the three waves of transmission in a small province in China. The data is presented neatly. However the conclusions do not fully reflect the results obtained. English grammar, formation of sentences, spelling errors and syntaxal errors abound throughout the manuscript and need very urgent attention

Author response: Dear Reviewers, Thank you very much for your valuable suggestions on this study. We have made a number of revisions to the results and conclusions of this paper. We have added spatial autocorrelations for the three phases of the aggregated epidemic in Jilin Province to understand the spatial and temporal distribution of COVID-19 in these three phases, which has explained the phenomenon in its epidemiology. We have made significant revisions. In the methods section, we added methods for analysing global and local spatial autocorrelations of the incidence of the three stages of the aggregated epidemic in Jilin Province; in the results section, we added analysis of spatial heterogeneity across regions in the three stages of the epidemic in Jilin Province. Finally, we have deep dug into all the results sections and reworked the discussion section. For details see lines 206 and 218, 347 and 354, and 466 and 476.

Finally, we have reached out to the English editors at Editage (www.editage.cn) for assistance and have made significant revisions to the language of this article.

---

## [Decision Letter · Decision Letter 1]

19 Dec 2022

Epidemiological Clustered characteristics of coronavirus disease 2019 (COVID-19) in three phases of transmission in Jilin Province, China

PONE-D-21-36962R1

Dear Dr. Yu,

We’re pleased to inform you that your manuscript has been judged scientifically suitable for publication and will be formally accepted for publication once it meets all outstanding technical requirements.

Kind regards,

Academic Editor

PLOS ONE

Additional Editor Comments (optional):

Reviewers' comments:

Reviewer's Responses to Questions

**Comments to the Author**

1. If the authors have adequately addressed your comments raised in a previous round of review and you feel that this manuscript is now acceptable for publication, you may indicate that here to bypass the “Comments to the Author” section, enter your conflict of interest statement in the “Confidential to Editor” section, and submit your "Accept" recommendation.

Reviewer #1: All comments have been addressed

2. Is the manuscript technically sound, and do the data support the conclusions?

Reviewer #1: Yes

3. Has the statistical analysis been performed appropriately and rigorously? 

Reviewer #1: Yes

4. Have the authors made all data underlying the findings in their manuscript fully available?

Reviewer #1: Yes

5. Is the manuscript presented in an intelligible fashion and written in standard English?

Reviewer #1: Yes

6. Review Comments to the Author

Reviewer #1: The topic is interesting and the authors have addressed my concerns in the section of limitation of the study.

7. PLOS authors have the option to publish the peer review history of their article (what does this mean?). If published, this will include your full peer review and any attached files.

Reviewer #1: No

---

## [Editor Report · Acceptance letter]

9 Jan 2023

PONE-D-21-36962R1 

Epidemiological clustered characteristics of coronavirus disease 2019 (COVID-19) in three phases of transmission in Jilin Province, China 

Dear Dr. Yu:

I'm pleased to inform you that your manuscript has been deemed suitable for publication in PLOS ONE. Congratulations! Your manuscript is now with our production department. 

Kind regards, 

on behalf of

Dr. Robert Jeenchen Chen 

Academic Editor

PLOS ONE